# Single caudate neurons encode temporally discounted value for formulating motivation for action

**Yukiko Hori[1], Koki Mimura[1], Yuji Nagai[1], Atsushi Fujimoto[1], Kei Oyama[1], Erika Kikuchi[1], Ken-ichi Inoue[2], Masahiko Takada[2], Tetsuya Suhara[1], Barry J Richmond[3], Takafumi Minamimoto[1]***

[1]Department of Functional Brain Imaging, National Institutes for Quantum and Radiological Science and Technology, Chiba, Japan; [2]Systems Neuroscience Section, Primate Research Institute, Kyoto University, Inuyama, Japan; [3]Laboratory of Neuropsychology, National Institute of Mental Health, National Institutes of Health, Department of Health and Human Services, Bethesda, United States

**Abstract** The term 'temporal discounting' describes both choice preferences and motivation for delayed rewards. Here we show that neuronal activity in the dorsal part of the primate caudate head (dCDh) signals the temporally discounted value needed to compute the motivation for delayed rewards. Macaque monkeys performed an instrumental task, in which visual cues indicated the forthcoming size and delay duration before reward. Single dCDh neurons represented the temporally discounted value without reflecting changes in the animal's physiological state. Bilateral pharmacological or chemogenetic inactivation of dCDh markedly distorted the normal task performance based on the integration of reward size and delay, but did not affect the task performance for different reward sizes without delay. These results suggest that dCDh is involved in encoding the integrated multi-dimensional information critical for motivation.

*For correspondence:
minamimoto.takafumi@qst.go.jp

**Competing interests:** The authors declare that no competing interests exist.

## Introduction

Motivation for engaging in action depends on the expected value of its outcome, e.g., when and how much money or food will be available as a reward. Intuitively, the larger and earlier the reward is, the greater the motivation will be. When animals and humans suppose the reward to be delayed, their behaviors become slower and less accurate. This decline in motivation is conceptualized as discounting of reward value as a function of time, namely temporal discounting (*Minamimoto et al., 2009*; *Shadmehr et al., 2010*; *Berret and Jean, 2016*). Temporal discounting was originally proposed to describe choice preferences for earlier smaller rewards rather than later larger rewards (*Mazur, 1984*; *Mazur, 2001*; *Green and Myerson, 2004*), implying that motivation and decision-making may share common brain processes. Besides temporal discounting, motivational processes also consider internal drive for reward, such as hunger and thirst, integrating these two factors into motivational value (*Toates, 1986*; *Berridge, 2004*; *Zhang et al., 2009*).

One of the major candidates as the neural systems mediating the computation of expected outcome value and transforming it into action is the basal ganglia (*Daw and Doya, 2006*; *Hikosaka et al., 2006*). Several lines of evidence based on electrophysiological studies have suggested that the caudate nucleus (CD) plays an important role in motivational processing via signaling an expected outcome and monitoring action/outcome leading to future behavioral improvement (*Kawagoe et al., 1998*; *Cromwell and Schultz, 2003*; *Lau and Glimcher, 2008*; *Hori et al., 2009*). Especially, the dorsal part of the head of the CD (dCDh) is best situated to participate in temporal discounting processes because it receives strong convergent inputs from various frontal cortical

areas including the dorsolateral prefrontal cortex (DLPFC), the anterior cingulate cortex (ACC), and the supplementary eye field (SEF) (*Haber et al., 1995*; *Haber et al., 2006*), where neuronal activity is related to the expected amount or delay/proximity of rewards (*Shidara and Richmond, 2002*; *Roesch and Olson, 2003*; *Roesch and Olson, 2005*; *Tsujimoto and Sawaguchi, 2005*; *Sohn and Lee, 2007*; *So and Stuphorn, 2010*). Indeed, it has been shown that neurons in this CD sector respond in relation to temporally discounted values during intertemporal choice (*Cai et al., 2011*). However, it is not yet clear how dCDh contributes to the computation of motivational value with temporal discounting.

Here, we examined single-unit activity in dCDh of macaque monkeys while they performed a delayed reward task. In the task a visual cue indicated the forthcoming reward size and the delay duration to the reward after simple action. From each animal's behavior, we were able to infer the value for temporally discounted rewards including their interactions with satiation. We found that a subpopulation of single dCDh neurons increased their activity during the time period from the cue onset to the execution of action. The activity of many neurons was correlated with the temporally discounted value related to the expected value of outcome. However, the activity was not influenced by the level of satiation. To determine whether the value-related activity might be causally related to behavior, pharmacological inactivation (local muscimol injection) and chemogenetic inactivation (designer receptor exclusively activated by designer drugs, DREADDs) (*Nagai et al., 2016*; *Roth, 2016*) of dCDh were carried out; both of these inactivations produced consistent impairments in motivational behaviors reflected as a distorted integration of reward size and delay, while behaviors based on the integration of reward size and physiological state remained intact.

## Results

### Temporal discounting accounts for monkeys' behavior

We studied computation of the motivational value using temporal discounting in macaque monkeys induced delaying reward delivery (*Figure 1A*). In the basic task, the monkey must release a bar when a red spot turns green. A visual cue appears at the beginning of each trial and remains on throughout. Each of the six cues is linked to one combination of reward size (one small drop; three or four large drops) and delay to reward (one of 0, 3.3, and 6.9 s; *Figure 1B*). In this and similar tasks, the error rate, i.e., the proportion of trials with an incorrect response (either releasing the bar too early or too late), reflects the monkey's motivation for action, which can be interpreted as the motivational value or decision utility for whether to act or not, according to its prediction about the forthcoming reward. In our previous studies, the error rate was inversely related to the motivational value (*Minamimoto et al., 2009*). In previous behavioral studies, the subjective value of delayed reward was formulated as a hyperbolic discounting model (*Mazur, 1984*; *Mazur, 2001*; *Green and Myerson, 2004*),

$$DV = \frac{R}{1 + kD} \tag{1}$$

where *DV* is the value of delayed reward (i.e., temporally discounted value), *R* is the magnitude of reward, *k* is a discount parameter, and *D* is the delay to the reward. Accordingly, to describe error rates in this delayed reward task, we extended the inverse relation, incorporating it into a hyperbolic discounting model as shown in *Equation 2*, with error rates (*E*), reward size (*R*), delay (*D*), and a monkey-specific free-fitting parameter (*a*) (*Minamimoto et al., 2009*),

$$E = \frac{1 + kD}{aR} \tag{2}$$

As shown in *Figure 1C*, the error rates were higher when a small reward size was expected, and for both reward sizes, the errors increased linearly as the expected delay duration increased. This pattern of the averaged error rates was well described by the inverse relation with hyperbolic delay discounting (*Equation 2*) ($R^2$ = 0.96, 0.88, and 0.94 for monkeys BI, FG, and ST, respectively; *Figure 1C*, solid lines). The exponential discounting model (*Equation 3*) also explained the majority of the cases (7/10 monkeys, $R^2$ > 0.9; e.g., *Figure 1C*, dotted curves for monkeys BI and ST) well. Consistent with previous results (*Minamimoto et al., 2009*), leave-one-out cross-validation analysis

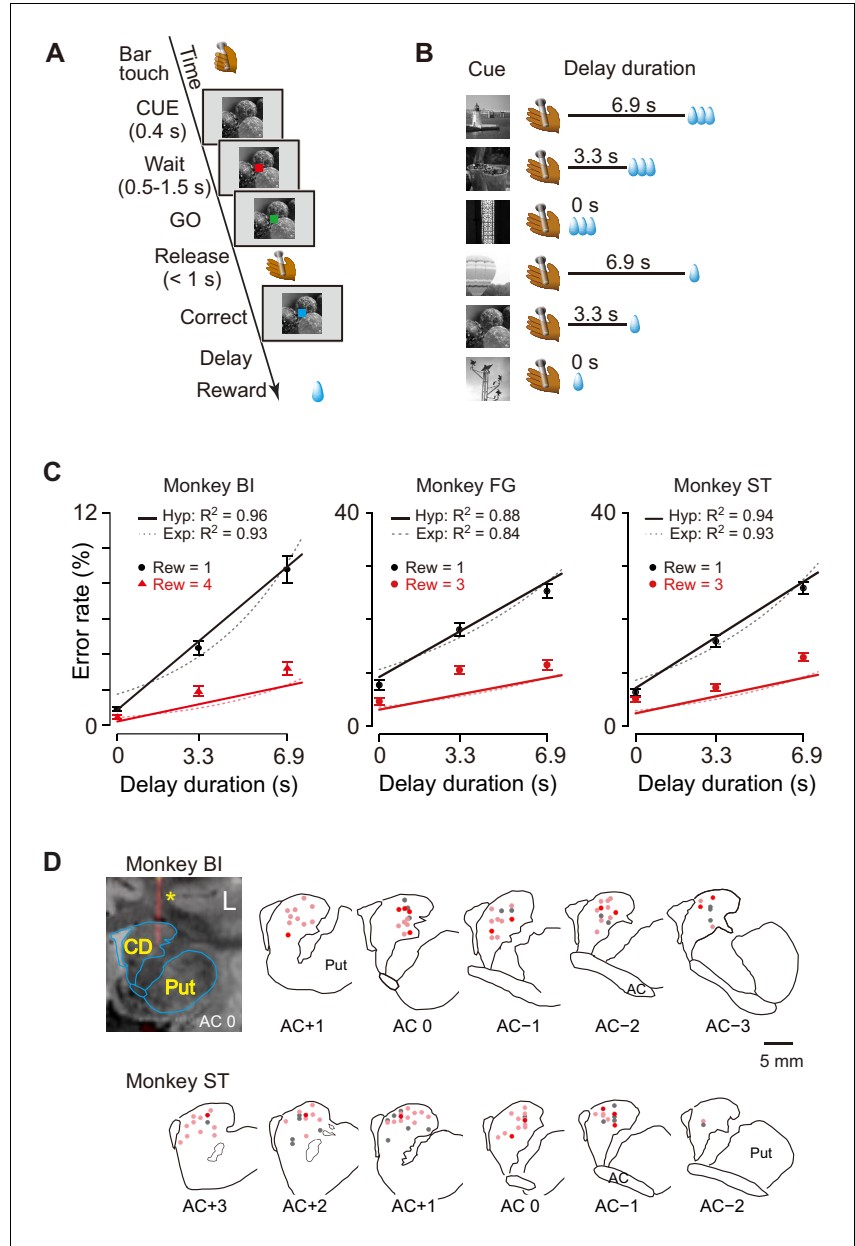

**Figure 1.** Task, behavioral performance, and recording sites. (**A**) Sequence of events of behavioral tasks. (**B**) Example of relationship between cue and outcome in delayed reward task. (**C**) Ratio of error trials (mean ± sem) as a function of delay duration in monkeys BI, FG, and ST. Data of small (one drop) and large reward (three or four drops) trials are indicated by black and red, respectively. Solid lines and dotted curves are best fit of *Equations 2 and 3*, respectively. Note that since two straight lines were simultaneously fitted to the averaged data, the fitting was worse for the data of trials with larger rewards. (**D**) Series of coronal sections illustrating locations of recorded neurons plotted by dots. Anterior–posterior positions of sections (distance, in mm) are indicated by plus and minus numbers from anterior commissure (AC), respectively. Red, cue-responsive neurons with DV coding; pink, cue-responsive neurons without DV coding; gray, neurons without cue response. Coronal sections of CT-MR fusion image in top left visualize an electrode (*) in dCDh. CD, caudate nucleus; Put, putamen.

The online version of this article includes the following source data and figure supplement(s) for figure 1:

**Source data 1.** Souce data of error rates as a function of delay duration and reward size.
**Figure supplement 1.** Error type and timing, and reaction time and eye position.
**Figure supplement 2.** Eye position during cue period.

confirmed that the hyperbolic model fitted the error rates significantly better than exponential function for all three monkeys as well as for seven additional monkeys (p<0.05; see Materials and methods).

The proportion of early errors differed across monkeys, but was relatively consistent within each monkey (*Figure 1—figure supplement 1A*). Nine of 10 monkeys exhibited a pattern in which early errors increased over time, reaching a peak at about 0.7 s or 1.8 s after cue onset, while only one monkey (monkey TM) showed an increase in early errors immediately after cue onset. These results suggest that early errors were not rejection responses, but rather the consequence of insufficient motivation to make the correct response. In addition, the late releases did not always occur immediately after the end of the 1 s response window, suggesting that they were not due to extensions of slow reaction (*Figure 1—figure supplement 1B*). These results also support the interpretation that errors are caused by insufficient motivation to respond correctly.

The reaction times also covaried with both reward size and reward delay; reaction times were shorter for larger rewards (two-way ANOVA; p<0.001, 8/10 monkeys including monkeys BI, FG, and ST) and shorter delays (p<0.001, 9/10 monkeys including monkeys BI, FG, and ST, *Figure 1—figure supplement 1C*). Although the monkeys were not required to fixate during the task, they usually gazed at the cue during the cue period. We did not find any significant effect of forthcoming reward size or delay duration on the duration of gazing at the cue (two-way ANOVA; main effect of reward size, effect size $\eta^2 = 0.004$; main effect of delay, $\eta^2 = 0.002$; reward size $\times$ delay, $\eta^2 = 0.003$) (*Figure 1—figure supplement 2*).

Together, these results suggest that the monkeys adjusted their motivation of action based on the temporally discounted value, which forms a hyperbolic relationship between expected size and delay of forthcoming reward.

## Neuronal activity of dCDh reflects temporally discounted value

We examined the role of the caudate nucleus in the motivational control of action based on the temporally discounted value. Specifically, we focused on dCDh and recorded the activity of 150 presumed projection neurons (i.e., phasically active neurons; see Materials and methods) (*Figure 1D*), while the monkeys performed the delayed reward task. Most of the neurons (n = 118) significantly increased their activity around more than one of three task phases: *cue* (immediately after cue appearance), *release* (at the time of bar release), and/or *reward* (at the time of reward delivery) (*Figure 2A–C*; p<0.05, $\chi^2$ test). The cue response was the most prominent activity of dCDh neurons during the task (*Figure 2A,C*); the proportion of cue-responsive neurons (100/150) was significantly larger than that of release-responsive neurons (49/150; p<0.01, $\chi^2$ test) and reward-responsive neurons (*Figure 2D*; 49/150; p<0.0001, $\chi^2$ test).

Some of the cue responses signaled a temporally discounted value (DV) of the forthcoming reward (*Equation 1*). An example of the activity shown in *Figure 3A* exhibited the strongest activation after the cue associated with a large and immediate reward. The cue response became smaller as the delay duration became longer, and with the smallest reward with long delay, the neuron did not respond at all. The neuron presented in *Figure 3B* had the opposite response pattern; the activation was stronger when the cue predicted smaller rewards with longer delays.

We related spike discharge rates to DV estimated using the hyperbolic function obtained from individual behavior (*Equations 2 and 7*). The firing rate during the cue period of example neurons (*Figure 3A,B*) correlated with DV positively (*Figure 3C*, $R^2 = 0.86$, p<0.01) and negatively (*Figure 3D*, $R^2 = 0.77$, p<0.05). A significant regression coefficient for DV (p<0.05, t-test) was found in 27 of 100 cue-responsive neurons (11, 6, and 10 in monkeys BI, FG, and ST, respectively); 18 and 9 exhibited positive and negative correlations, respectively. The result did not seem to depend on the shape of DV function: a similar number of neurons showed a significant DV relation when estimating using the exponential function (*Equation 3*; n = 25). By contrast, significant DV relation was relatively minor in release-related (5/49) and reward responses (3/49). The DV relation was not likely to be a direct reflection of the eye movement or gaze variables, since the monkeys tended to looked at cue location from cue to go signal regardless of rewarding condition (*Figure 1—figure supplement 2*).

Besides the DV relation, the cue response might solely reflect reward size or delay duration. We compared the effect of size or delay alone on cue response with that of DV using multiple linear regression analysis (*Equation 8*). We found that only three and four neurons showed a significant

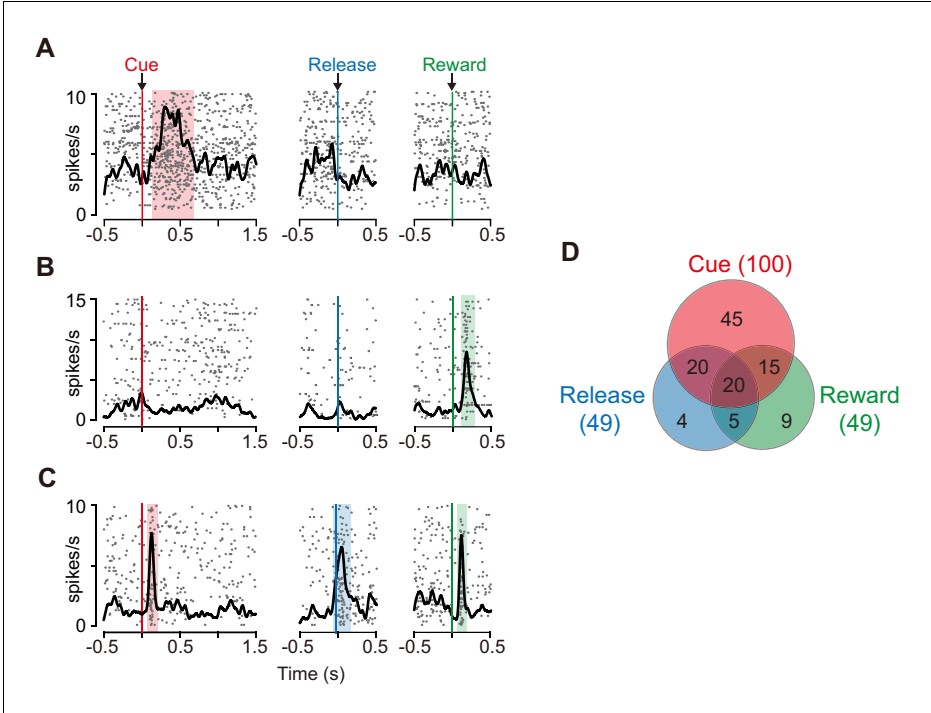

**Figure 2.** Task-related responses of dCDh neurons. (**A**) Example of a neuron that responded exclusively to cue. Rasters and spike density histograms for all trials are aligned at the cue signal (left), bar release (middle), and reward delivery (right). Rasters are shown in order of occurrence of trials from bottom to top. Shaded areas are time windows when discharge probability is significantly higher than baseline (p<0.05, $\chi^2$ test). (**B**) Example of a neuron that responded exclusively to reward delivery. (**C**) Example of a neuron that responded to cue, bar release and reward delivery. (**D**) Distribution of neurons that responded in three task phases shown in Venn diagram. Numbers in parentheses represent numbers of neurons showing significant response to each event. The proportions of responded neurons in each monkey are as follows: Cue, 88%, 88%, and 83%; Release, 37%, 42%, and 46%; Reward, 41%, 50%, 38%; for monkeys BI, FG, and ST, respectively.

exclusive effect of size or delay on their cue response, respectively (**Figure 4A,B**, blue and green). In contrast, for 19 and 5 neurons, DV and both delay and size had a significant effect on the cue response, respectively (**Figure 4A,B**, red and pink), the proportions of which were significantly larger than that of neurons by chance coding both delay and reward size (p<0.01; $\chi^2$ test). The strength of size or delay effect was relatively smaller than that of DV. Thus, DV-related neurons were not just a selected population from the neurons representing mixtures of these delay and size by chance; rather, the entire neuronal population seemed to represent reward size and delay in an integrated manner. Such population-level DV relation was also observed in the release response, but not in the reward response (**Figure 4—figure supplement 1**).

Together, our results suggest that the temporally discounted value of the forthcoming reward is represented in dCDh neurons, that is, mainly in a subpopulation of neurons. In the following section, we will focus on this subset of neurons and refer to neurons with and without significant correlation with DV as *DV-coding neurons* (n = 27) and *non-DV-coding neurons* (n = 73), respectively. DV-coding neurons were not confined to specific locations, but were found throughout the dCDh (**Figure 1D**).

To quantify the time course of DV coding of the cue responses, the effect size of DV ($R^2$) in a linear regression analysis (**Equation 7**) was calculated (200 ms window, 10 ms steps) for each DV-coding neuron (**Figure 5A,B**). On average, the effect size rose from 100 ms after cue onset, reaching a peak at 750 ms after the cue (red curve, **Figure 5C**). Thereafter, it gradually decreased to the bar release (**Figure 5D**). The effect size did not become 0, indicating that a few neurons (n = 5) also signaled DV around bar release. Thus, DV coding started just after the monkey was informed about the reward size and delay of the forthcoming reward, and it continued until the time point of execution of an action. We postulated that the activity of DV-coding neurons may be related to the process

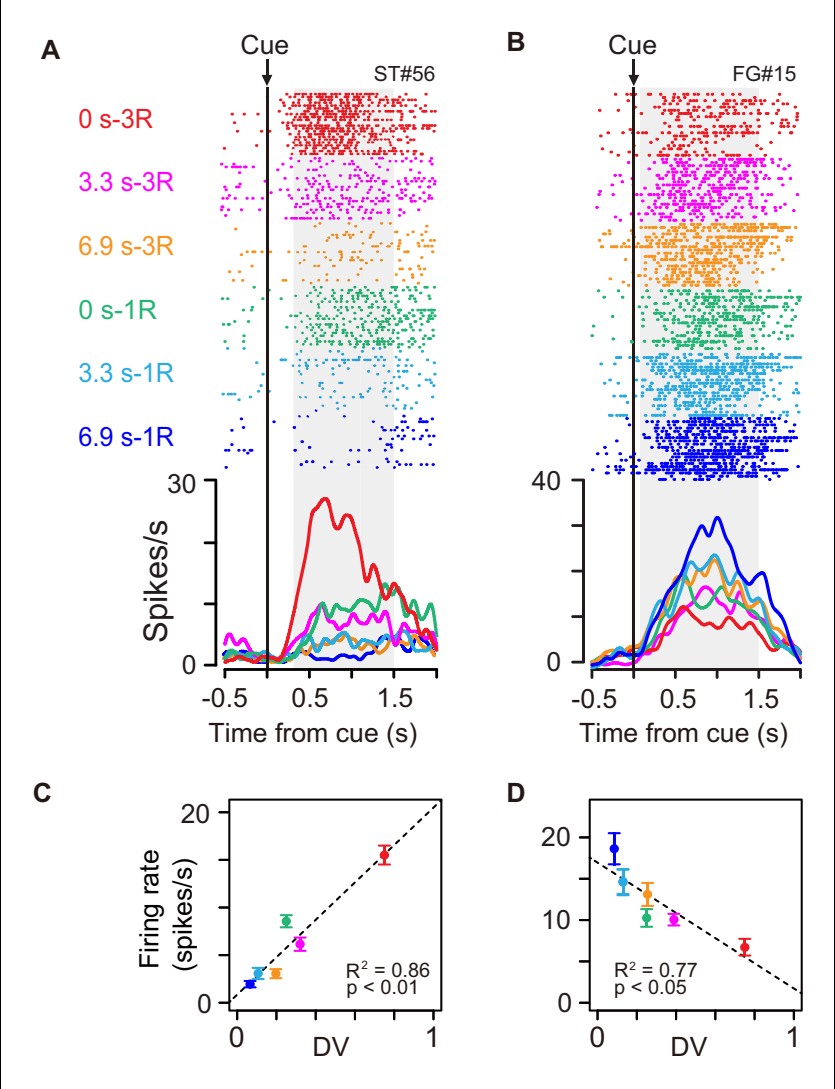

**Figure 3.** Cue responses of temporally discounted value coding. (**A**, **B**) Activity of example neurons during cue period. Rasters and spike density histograms are aligned at cue onset. The color corresponds to each reward condition. Rasters are shown in order of occurrence of trials from bottom to top in each condition. Shaded areas on rasters are time windows for evaluating the magnitude of cue response. (**C**, **D**) Relationship between firing rate (mean ± sem) and temporally discounted value (DV, *Equation 1*) for neuronal activities shown in (**A**) and (**B**), respectively.

The online version of this article includes the following figure supplement(s) for figure 3:

**Figure supplement 1.** Error trial analysis.

**Figure supplement 2.** Error trial analysis.

mediating outcome prediction and further the decision to act or not. If this is the case, the DV-coding neurons should behave differentially between correct and error trials. To test this, we performed linear mixed model (LMM) analysis on 22 of 27 DV-coding neurons recorded in a session in which the monkeys made at least three error trials. We found that the majority of DV-coding neurons (17 of 22) were modulated differentially by DV depending on whether the monkey performed correctly or not (*Figure 3—figure supplements 1* and *2*), supporting the idea that this population of neurons is involved in motivational processes.

Non-DV-coding neurons, on the other hand, did not change the effect size from 0 during the cue period, whereas it increased after the bar release (black curve, *Figure 5C,D*). Comparing the normalized activity of these two populations, whereas DV-coding neurons showed an increase in activity

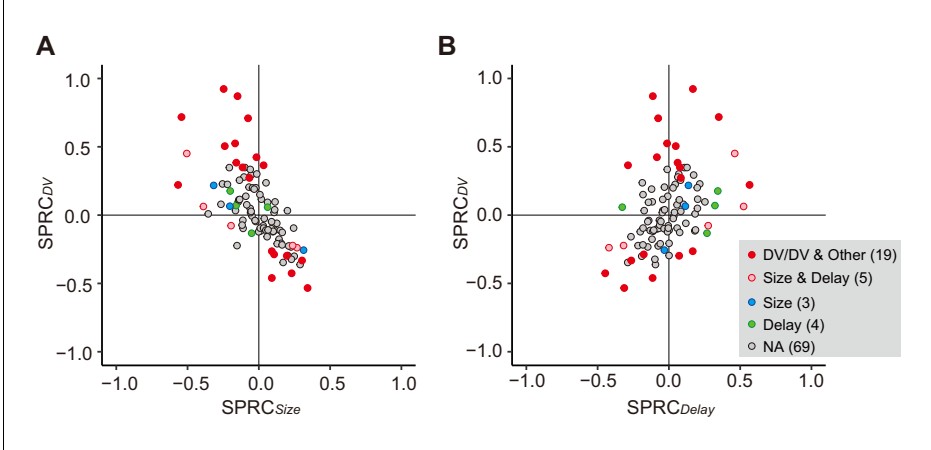

**Figure 4.** Impact of DV and comparison with delay and size on cue response. (A, B) Scatterplots of standardized partial regression coefficients (SPRC) of DV (ordinate) against those for reward size or delay (abscissa) for discharge rates during cue period, respectively. Colored dots indicate neurons with significant (p<0.05) coefficient, while gray dots correspond to neurons without any significant effect (NA). DV/DV and Other, neurons with significant coefficient of DV; Size and Delay, those with both size and delay; Size, those exclusively with size; Delay, those exclusively with delay. Numbers in parentheses indicate number of neurons.

The online version of this article includes the following figure supplement(s) for figure 4:

**Figure supplement 1.** Impact of DV and comparison with delay and size on release and reward response.

toward the bar release, non-DV-coding neurons showed a marked transient response to the cue (*Figure 5E,F*). Given that the monkeys tended to look at the cue location during cue period (*Figure 1—figure supplement 2B*), the activity of non-DV-coding neurons appeared to largely reflect visual response, but was unlikely to be evoked by eye movement. This suggests that non-DV-coding neurons might have a role in detecting cue appearance.

## DV coding is insensitive to satiation

The motivational value of reward should decrease as the physiological drive state changes from thirst to satiation. In every daily session, the monkeys were allowed to work until they stopped by themselves, meaning that the data were collected as the monkeys were approaching satiation. As the normalized cumulative reward ($R_{cum}$) increased, the overall error rate in each combination of reward size and delay also increased (*Figure 6A*). When we looked at the data from one quarter (e.g., *Figure 6B*, $R_{cum}$ = 0.75–1), the error rate increased linearly as the delay duration increased with each reward size. These observations were well in accordance with the psychological concepts of incentive motivation assuming a multiplicative interaction between the value of outcome (i.e., discounted value) and the satiation effect (*Toates, 1986*; *Berridge, 2004*; *Zhang et al., 2009*; *Equation 6*; see Materials and methods). The error rates were well explained by *Equation 4* for each individual monkey ($R^2$ = 0.89 ± 0.06; mean ± sem) as well as for the average across nine monkeys ($R^2$ = 0.98, *Figure 6A,B*). The satiation effect, $F(R_{cum})$ (*Figure 6C*), indicated that the motivational value of reward decreased at a rate of more than 15% (16%, 33%, and 17% for BI, FG, and ST, respectively) in a single recording session (i.e., 120 trials) according to the number of average success trials in a daily session.

Although satiation significantly influenced behavior, satiation did not influence dCDh activity, not even when coding DV. When we compared the cue responses between the first and second halves of 120 successful trials, the activity patterns were indistinguishable between the first and second halves of the recording period in a single neuron (*Figure 6D*). Similarly, the normalized mean discharge rate of cue responses for each reward condition did not significantly change between the first and second halves in 18 positive DV-coding neurons (repeated-measures two-way ANOVA; main effect of trial type, $F_{(5, 119)}$ = 16.8, $p<10^{-8}$; main effect of satiation, $F_{(1, 119)}$ = 1.7, p=0.29; *Figure 6E*). Additional neuron-by-neuron analysis using a multiple linear regression model (*Equation 9*) demonstrated that a significant satiation effect was not found in any of the cue-responsive

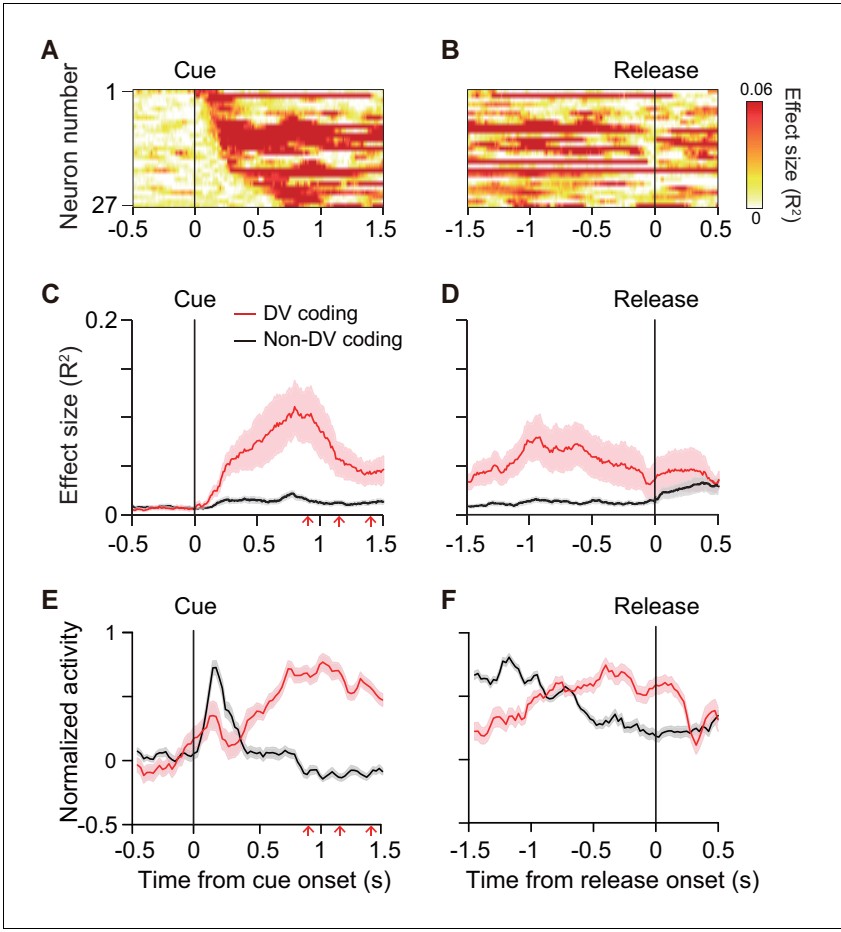

**Figure 5.** Time course of DV coding. (**A, B**) Time-dependent change of DV coding. Each row represents color-coded effect size ($R^2$) of DV for a single DV-coding neuron. Responses were aligned by cue onset and bar release, respectively. (**C, D**) Time-dependent change of effect size of DV for DV coding (red, n = 27) and non-DV-coding neurons (black, n = 73) aligned by cue onset and bar release, respectively. Thick curve and shaded areas indicate mean ± sem, respectively. Arrows indicate time of go signal (first 3 of 5 with variable interval). (**E, F**) Time course of normalized activity for DV coding (red, n = 27) and non-DV-coding neurons (black, n = 73) aligned by cue onset and bar release, respectively. Conventions are the same as (**C, D**).

The online version of this article includes the following source data for figure 5:

**Source data 1.** Source data of time-dependent change of DV coding in individual neurons.

dCDh neurons (97/100) except for three non-DV-coding neurons (*Figure 6—figure supplement 1*). Therefore, dCDh neurons encode the expected temporally discounted value in their cue response without reflecting internal physiological drive.

## Inactivation of dCDh specifically impairs behavioral pattern to delay discounting

In our results, the activity of a subset of dCDh neurons encoded DV after the cue, but not reward size or delay alone. This raises the question of whether the activity is needed to judge the values reflected by DV. To test this, we inactivated bilateral dCDh by local injection of muscimol (GABA_A receptor agonist) or by a chemogenetic technology (DREADDs), two complementary methods to produce the comparable behavioral change when applied to the primate striatum (*Nagai et al., 2016*). Two monkeys had muscimol injected locally into the dCDh, which was confirmed by CT images of injection cannulae overlaying MR images, matching with the recording sites (*Figure 7A and B*; see *Figure 1D* for comparison). Another monkey received injections of a viral vector expressing an inhibitory DREADD, hM4Di, into the dCDh bilaterally. A positron emission tomography (PET)

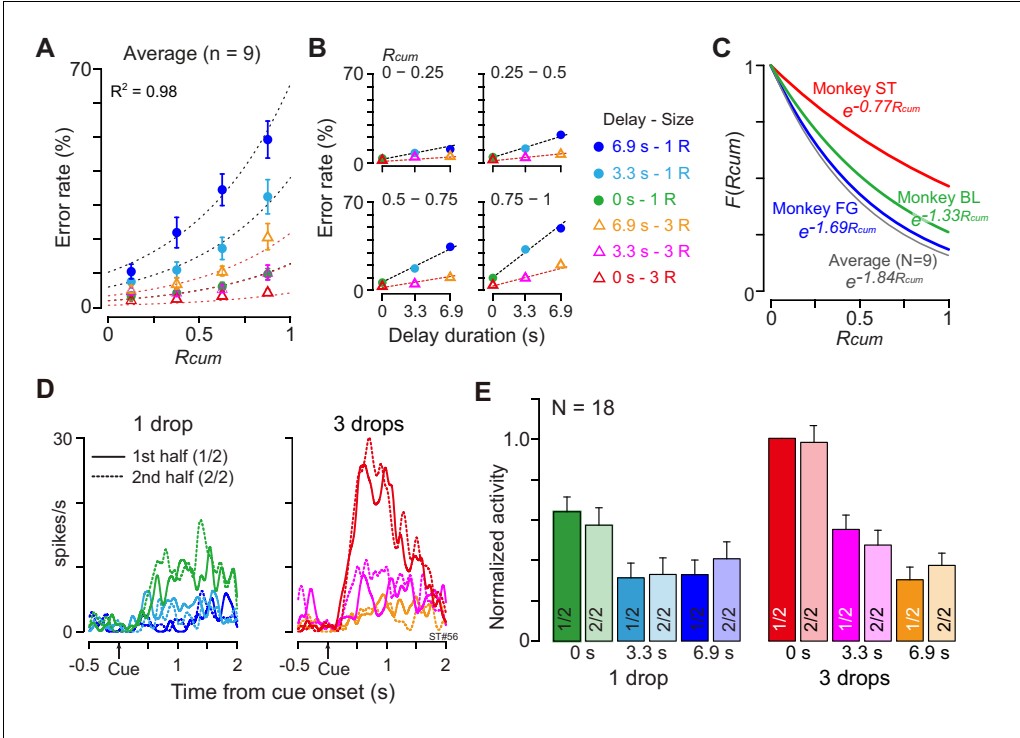

**Figure 6.** Negligible effect of satiation on DV-coding. (**A**) Ratio of error trials (mean ± sem) as a function of normalized cumulative reward ($R_{cum}$) on average across nine monkeys. Dotted curves are the best fit of *Equation 4* to the data. (**B**) Error rates (mean ± sem) as a function of delay duration for each quarter of $R_{cum}$. (**C**) Satiation function, $F(R_{cum})$, along with $R_{cum}$ in three individual monkeys and average across nine monkeys. Since average total trials were 934, 512, and 493 in BI, FG, and ST, motivational value became 84%, 67%, and 83% through 120 trials (i.e., 16%, 33%, and 17% devalued), respectively. (**D**) Example of comparison of cue responses in first and second half of recording period for each reward condition in single dCDh neuron (monkey ST). Spike density histograms are aligned at cue onset; one and three drops in reward size, respectively. (**E**) Comparison of cue responses in first and second half of recording period for each trial type in positive DV-coding neurons (n = 18). Responses were normalized by firing rate of cue response in immediate large reward trials during first half of the period.

The online version of this article includes the following figure supplement(s) for figure 6:

**Figure supplement 1.** Impact of discounted value and satiation on cue response.

scan with a DREADD-selective radioligand, [$^{11}$C]deschloroclozapine (DCZ) (*Nagai et al., 2020*), confirmed that hM4Di expression covered the dCDh (*Figure 7C*). Chemogenetic silencing was achieved by systemic administration of the selective DREADD agonist DCZ (*Nagai et al., 2020*). Both pharmacological and chemogenetic inactivation resulted in a significant shift in error rate patterns with respect to reward size and delay in all three monkeys (*Figure 7D*, left); the behavioral patterns were idiosyncratic across the monkeys, but they were generally not in accordance with the temporal discounting model (i.e., *Equation 2*; $R^2$ = 0.41, 0.19, and 0.76, for monkeys BI, RI, and ST, respectively). By contrast, the error rate pattern following vehicle injection remained well explained by the model (*Figure 7D*, right; $R^2 > 0.86$).

Despite changing error patterns, inactivation did not produce statistically significant changes in the overall error rates (inactivation vs. control; two-way ANOVA for treatment × reward condition; main effect of treatment, $F_{(1,\ 2)}$ = 13.6, p=0.07; interaction, $F_{(5,\ 164)}$ = 2.1, p=0.07). Apart from the error rates, the inactivation did not affect other behavioral parameters. The total reward earned during the task was unchanged in each monkey (inactivation vs. control; Brunner–Munzel test, p>0.18). There was no significant effect of treatment on reaction time in two monkeys (two-way ANOVA, effect size of treatment: $\eta^2 < 0.01$, monkeys BI and RI) but a moderate effect of treatment in monkey ST ($\eta^2$ = 0.06) (*Figure 7—figure supplement 1*). Type of error (i.e., releasing too early or too late)

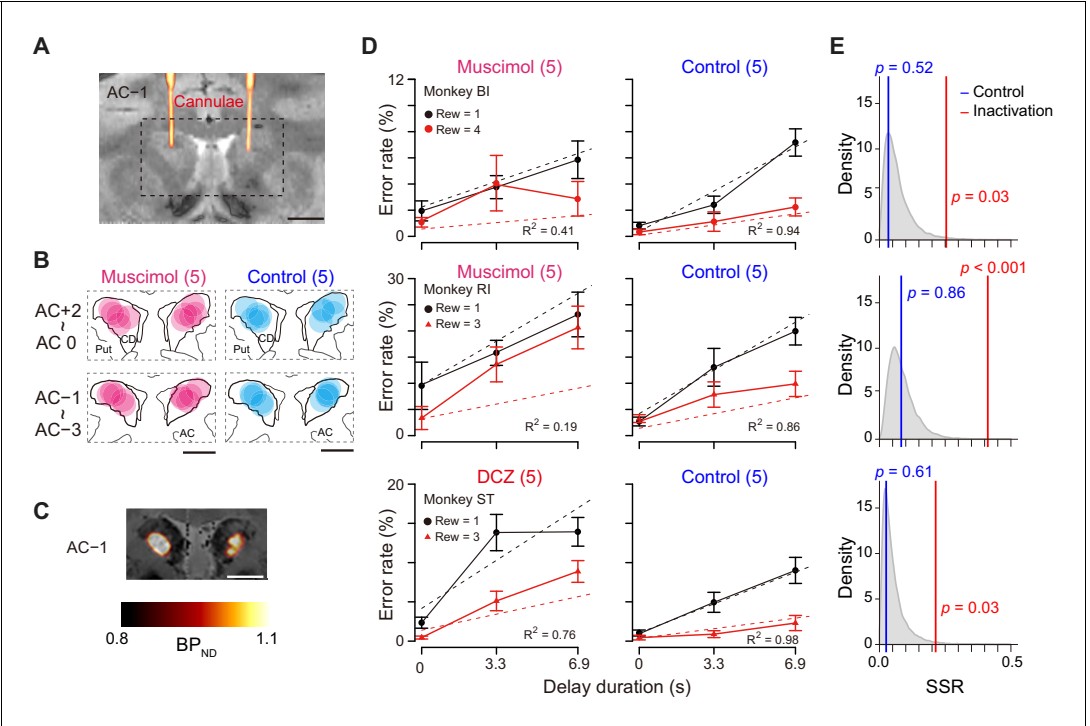

**Figure 7.** Bilateral inactivation of dCDh disrupted normal motivational performance based on size and delay. (A) CT-based localization of muscimol injection sites. CT image visualizing injection cannulae targeting CD bilaterally (hot color) overlaid on MR image (gray scale) in monkey BI. Scale bar, 5 mm. (B) Muscimol (magenta) and saline injection sites (blue) are mapped by estimating diffusion (4 mm in diameter) from the tip of the cannula. The data of two subjects are overlaid and are separately mapped 3 mm anterior and 3 mm posterior to the anterior commissure (AC). (C) [$^{11}$C]DCZ-PET visualizing hM4Di expression in vivo in monkey ST. Parametric image of specific binding (BP$_{ND}$) of [$^{11}$C]DCZ-PET overlaying MR image. Scale bar, 5 mm. (D) Error rates (mean ± sem) as function of delay duration under inactivation (left) and control condition (right). Black and red symbols are low and high reward trials, respectively. Dotted lines represent best-fit function of hyperbolic temporal discounting (*Equation 2*). Number in parentheses indicates number of sessions tested. (E) Distribution of sum of squared residuals (SSR) of best-fit function (*Equation 2*) to averaged resample data obtained by bootstrap method (n = 20,000). Blue and red lines indicate SSR of best fit of *Equation 2* to mean error rates in control and inactivation sessions, respectively.

The online version of this article includes the following source data and figure supplement(s) for figure 7:

**Source data 1.** Source data of error rates in control and inactivation sessions.
**Figure supplement 1.** No significant effects of dCDh inactivation on reaction time in delayed reward task.
**Figure supplement 2.** No effect of dCDh inactivation on eye position.
**Figure supplement 3.** Normalized error rates in baseline, control, and inactivation session of delayed reward task.
**Figure supplement 4.** Effect of dCDh inactivation on satiation.

was unaffected by inactivation (main effect of treatment, $F_{(1,26)}$ = 1.07, p=0.31). The monkeys touched and released the bar several times during the delay period, even though the delay time was not shortened. The number of releases depended on reward condition (main effect of reward condition, $F_{(5, 143)}$ = 25.22, p<0.001), but there was no significant main effect of treatment (two-way ANOVA, treatment, $F_{(1, 143)}$ = 2.90, p=0.09) or interaction ($F_{(5, 143)}$ = 0.42, p=0.83). The duration of gazing at the cue was slightly but not significantly longer during muscimol inactivation (t-test, p=0.063, *Figure 7—figure supplement 2*). Together, the bilateral inactivation of dCDh did not cause impairments in overall motivation, motor, or anticipatory behavior.

These results demonstrated that dCDh inactivation appeared to produce a consistent impairment, namely, alteration of error rate pattern without changing overall error rates. To quantify behavioral deviation from normal temporal discounting, we normalized the error rates in each session for baseline, inactivation, and control condition (*Figure 7—figure supplement 3*). Bootstrap analysis revealed that, compared to baseline data, inactivation, but not control, caused significant deviations in the error rate patterns away from the temporal discounting model in all monkeys

(p<0.05, *Figure 7E*, red line), suggesting that dCDh silencing distorted normal motivational value processing based on the integration between reward size and delay.

To examine the effect of dCDh inactivation on satiation, we plotted error rates along with the normalized cumulative reward ($R_{cum}$). Like the results shown in *Figure 6A*, the error rate in each combination of reward size and delay increased as $R_{cum}$ increased in baseline and vehicle control sessions (*Figure 7—figure supplement 4*). Satiation-dependent increase in error rates was also observed in two of three monkeys in dCDh inactivation, while monkey ST failed to show this tendency (*Figure 7—figure supplement 4*). We also examined trial initiation time (duration between the time the reward was received and the start of the next trial), reflecting satiation effects as a measure of motivation to start time in a previous study (*Fujimoto et al., 2019*). In both control and inactivation sessions, the trial initiation time was significantly longer in the second half of the session (two-way ANOVA, main effect of first vs second, $F_{(1,54)}$ = 4.32, p=0.042), where no significant interactive effect of dCDh inactivation was observed (first vs second × treatment, $F_{(1,54)}$ = 0.32, p=0.57). These results suggest that dCDh inactivation does not have a strong effect on satiation.

Was the impairment specifically related to the temporally discounted value? Alternatively, it may reflect the dysfunction of the motivational process in general. Since the temporally discounted value is often referred to as 'subjective value', dCDh inactivation could produce a general dysregulation of computation for motivational value – a subjective impact of the upcoming reward on performance. To examine the effects of dCDh inactivation on motivational value without delay, we tested two monkeys in a reward-size task in which the task requirement remained the same as the delayed reward task, but a successful bar release was immediately rewarded with one of four reward sizes

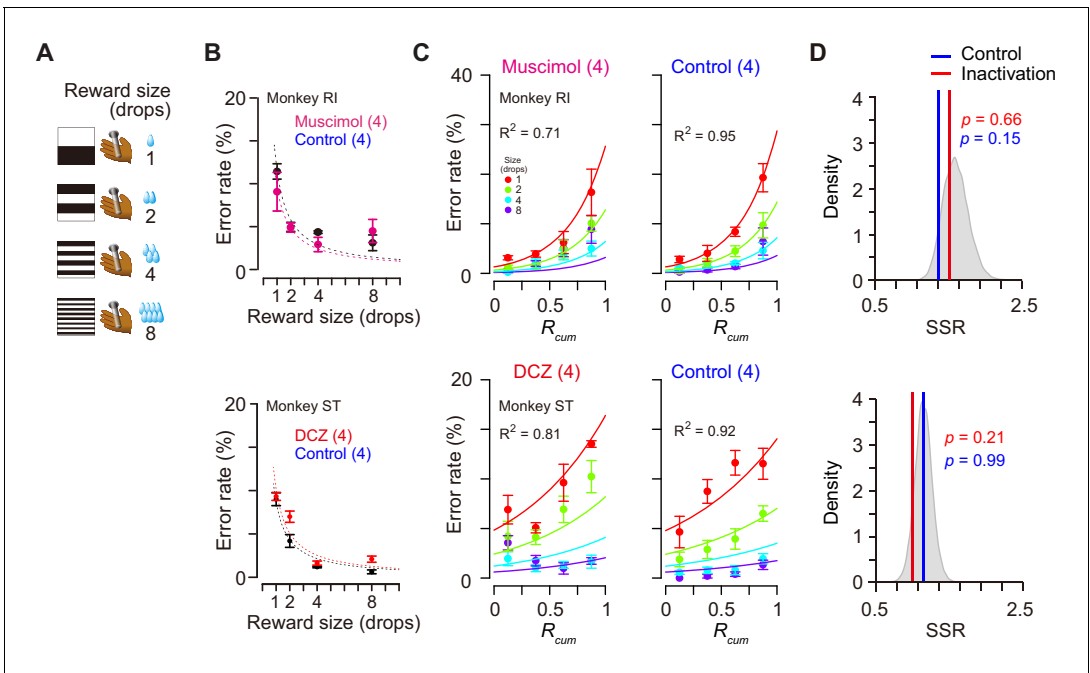

**Figure 8.** Reward-size task and behavioral performance. (**A**) Cue stimuli used in reward-size task uniquely associated with forthcoming reward size (one, two, four, or eight drops). (**B**) Top: Error rates (mean ± sem) as function of reward size in muscimol treatment (magenta) and non-treatment control session (black) for monkey RI, respectively. Bottom: Error rates (mean ± sem) as function of reward size after DCZ treatment (red) and after vehicle treatment (black) for monkey ST, respectively. Dotted curves represent best-fit of inverse function. (**C**) Error rates (mean ± sem) as function of normalized cumulative reward ($R_{cum}$) for monkeys RI (top) and ST (bottom), respectively. Each reward size condition was shown in a different color. Number in parentheses indicates numbers of sessions tested. (**D**) Distribution of sum of squared residuals (SSR) of best-fit function (*Equation 6*) to averaged resample data obtained by bootstrap method (n=20,000). Blue and red lines indicate SSR of best-fit of *Equation 6* to the mean error rates in control and inactivation sessions, respectively.

The online version of this article includes the following source data and figure supplement(s) for figure 8:

**Source data 1.** Source data of error rates in reward size task.
**Figure supplement 1.** Comparison of learning in reward size and delayed reward task.

(one, two, four, or eight drops) associated with a unique cue (*Figure 8A*). It has been repeatedly shown that the error rates of this task will be well explained by the joint function of reward size and satiation (*Equation 6*; *Minamimoto et al., 2009*; *Minamimoto et al., 2012b*; *Fujimoto et al., 2019*). Pharmacological or chemogenetic inactivation of bilateral dCDh did not alter the pattern of the error rate in this task; in both cases they remained to be well explained by the model ($R^2 > 0.7$) (*Figure 8C*) and were equally well compared with the baseline data ($p>0.15$, bootstrap significance test; *Figure 8D*). The inactivation did not change the overall error rates (three-way ANOVA, treatment, $F_{(1, 243)} = 1.35$, $p=0.45$) or the interactive effect with reward size on the error rates (treatment $\times$ size, $F_{(3, 243)} = 1.69$, $p=0.17$). The lack of change in the error rate pattern in the reward-size task could be attributed to the relative ease of associating cues with outcome compared to the delayed reward task. However, no clear difference was evident between the two tasks in establishing the cue–outcome relationship as judged by the behavior during the training period (*Figure 8—figure supplement 1*). Overall, these results suggest that dCDh activity is specifically involved in computing the motivational value based on delay discounting, rather than general motivational processes based on the integration of incentives and drive.

## Discussion

In the present study, we investigated the role of dCDh in formulating the motivational value of expected delayed rewards. The behavior showed that the likelihood of carrying out the trials for delayed rewards was well described by a model with hyperbolic discounting and satiation. There were two main findings. First, a substantial number of single dCDh neurons represented the temporally discounted values, combining the information about the reward size and delay in delivery. However, these same neurons did not reflect a decrease in internal physiological drive seen in the behavior as the monkeys became more satiated. Second, bilateral pharmacological or chemogenetic inactivation of dCDh distorted the motivational valuation derived from the integration of reward size and delay duration, whereas the relationship from the integration of reward size and physiological state remained intact. These results suggest a major contribution of dCDh in mediating the integrated external information that is critical for formulating the motivation for action.

Previous studies have suggested that the neuronal activity in the CD is involved in translating value into action by signaling multi-dimensional aspects of reward-related information, including presence/absence (*Kawagoe et al., 1998*; *Cromwell and Schultz, 2003*), probability (*Lau and Glimcher, 2008*; *Oyama et al., 2010*; *White and Monosov, 2016*), and size of reward (*Nakamura et al., 2012*; *Fujimoto et al., 2019*). Neurons in dCDh reflect the action values of a specific movement (*Samejima et al., 2005*; *Lau and Glimcher, 2008*) and might contribute to selecting an action that maximizes future rewards. In the present study, we found that the cue responses of a subpopulation of dCDh neurons reflected temporally discounted values that were inferred from the individual behaviors. It could not be a simple reflection of physical features of a visual cue, since the neuronal signal was observed irrespective of the cue sets used for assigning delayed reward, and since the neuronal correlates disappeared when the cue was randomized with respect to the outcome (data not shown). It has also been suggested that the basal ganglia are involved in assessing information processing for the duration of events or actions. Neuronal signals reflecting the duration of past events related to temporal discrimination were found in the anterior striatum including CD (*Chiba et al., 2008*). The CD neurons also showed ramping-up activity in response to stimuli that predict timing of action initiation (*Suzuki and Tanaka, 2019*). It might not be surprising that the neuronal signal reported here was related not only to the forthcoming reward timing, but also to the reward size, hence representing DV. Although it has been reported eye movement-related activity of CD neurons modulated by forthcoming rewarding conditions (*Watanabe et al., 2003*), the DV signal observed here could not be a direct reflection of eye movements or gaze variables, since the monkeys constantly looked at the cue during the cue period regardless of the rewarding condition (*Figure 1—figure supplement 2*).

The DV signal emerged just after cue onset, gradually increased, and then disappeared before execution of the action (*Figure 5*). This time course suggests that the neuronal signal does not simply convey the Pavlovian value of the cue, but can be related to the cognitive process mediating the outcome prediction underlying the decision of whether to act or not. This was supported by the results of the error trial analysis, which showed that most of the DV-coding neurons behaved

differently between correct and error (*Figure 3—figure supplement 1*). Compared with DV, the effect of the reward size or delay duration on cue responses was relatively weak (*Figure 4*), indicating that the signal integration may take place at least partially in some upstream brain area(s). The first plausible source of temporal discounting is a prefronto-striatal projection. Our recordings were carried out from dCDh, the region receiving direct input from the frontal cortical areas including DLPFC, ACC, and SEF (*Selemon and Goldman-Rakic, 1985*; *Calzavara et al., 2007*; *Averbeck et al., 2014*). DLPFC neurons encode DV as well as reward, delay duration, and target position during an intertemporal choice task (*Kalenscher et al., 2005*; *Kim et al., 2008*), exhibiting strong modulations in response to the delay combined with the amount of reward (*Tsujimoto and Sawaguchi, 2005*; *Sohn and Lee, 2007*; *Hosokawa et al., 2013*). The activity in ACC reflects the expected amount of reward (*Knutson et al., 2005*; *Amiez et al., 2006*) and the delay/proximity of rewards (*Shidara and Richmond, 2002*), as well as delay discounting for reward (*McClure et al., 2007*). Neurons in SEF are also modulated by the amount of reward and delay duration (*Roesch and Olson, 2003*; *Roesch and Olson, 2005*; *So and Stuphorn, 2010*). The second possible source is a nigrostriatal dopaminergic input. When a stimulus signaled the timing of reward delivery, the stimulus response of dopaminergic neurons declined hyperbolically with the delay duration (*Kobayashi and Schultz, 2008*). The third possible source is a thalamostriatal input arising from thalamic nuclei, including the centromedian-parafascicular (CM-Pf) complex (*Smith et al., 2004*). Neuronal activity in CM reflects the predicted outcome value (*Minamimoto et al., 2005*), but at this point, there is no evidence that it is involved in delay discounting.

Independent of temporal discounting, the motivational value of the cue should also decrease according to a shift in the internal physiological drive state. However, the effect of drive has been investigated separately from temporal discounting, and it has generally not been taken into account during studies of choice behavior. In our task, the changing motivational value was well approximated as being exponentially decreased along with reward accumulation (*Figure 6C*), while the relative effect of reward size and delay on decision appeared to be constant. This was in good agreement with psychological concepts of motivation, in which motivational value arises from a multiplicative interaction between external stimulus and physiological state (*Toates, 1986*; *Berridge, 2004*). This also suggests that temporal discounting and reward devaluation may be two independent processes, one exerting a hyperbolic effect of delay duration on the reward size changing in a trial-by-trial manner, and the other slowly decreasing the motivational value of the reward in response to reward accumulation. Our data support the notion that dCDh may be involved in the former process only; DV coding in dCDh was not sensitive to changes in internal drive (*Figure 6*, *Figure 6—figure supplement 1*). A similar insensitivity to satiation has been reported in terms of cue-related activity in the ventral striatum that was correlated with reward value (*Roesch et al., 2009*). This leaves an intriguing possibility, namely, that the insensitivity of internal drive may result from the motor output used; different data could be obtained if we tested monkeys with saccadic eye movements, in which neurons in dCDh are known to be involved. Satiety-dependent changes in neuronal activity have been seen in the orbitofrontal cortex (OFC), ventromedial prefrontal cortex (*Rolls et al., 1989*; *Critchley and Rolls, 1996*; *Bouret and Richmond, 2010*), rostromedial caudate nucleus (rmCD), and ventral pallidum (VP) (*Fujimoto et al., 2019*). Perhaps satiety-related signals would be represented in a network believed to be critical for guiding a choice of food based on internal drive (*Izquierdo and Murray, 2010*; *Murray and Rudebeck, 2013*). To formulate the motivational value for action, the physiological state or drive signal from this network may be integrated with temporal discounting in the basal ganglia-thalamocortical circuit, brain structures downstream from dCDh.

The causal contribution of DV coding in dCDh to temporal discounting was examined by pharmacological and chemogenetic inactivations, which are complementary and applicable to silencing primate striatal activity (*Nagai et al., 2016*). Muscimol inactivation is a standard procedure that has repeatedly been used in monkey studies. It has, however, major drawbacks: (1) the extent of an effective area is difficult to be controlled or identified (although we monitored the location of injection sites by computed tomography [CT]) and (2) when the experiments are repeated, mechanical damage to tissue would accumulate. The chemogenetic tool DREADDs, on the other hand, overcomes these problems; once a silencing DREADDs, hM4Di, is delivered, substantially the same neuronal population can be inactivated non-invasively and the effective region can be confirmed by PET imaging, as demonstrated here, and by traditional post-mortem histochemistry. We found that

inactivation of dCDh by either method produced consistent behavioral impairments; inactivation abolished the normal pattern of error rates derived from the integration of reward size and delay duration (*Figure 7*). This impairment cannot be explained simply by changes in the temporal discounting rate or alterations in the evaluation of single incentive factors. Our results are consistent with previous findings that both lesioning and inactivation of the dorsomedial striatum in rats, a homologue of dCDh in primates, reduced the sensitivity of instrumental performance to shifts in the outcome value (*Yin et al., 2005*; *Yin et al., 2008*). In contrast, dCDh inactivation did not impair motivation based on reward size alone or according to the integration of reward size and physiological state (i.e., motivational value; *Figure 8*). Thus, impairment can be attributed to the loss of DV coding seen in the activity of single dCDh neurons. Similar specific impairments have also been found in monkeys with bilateral ablation of DLPFC (*Simmons et al., 2010*). Given intact motivational evaluation for the reward size alone in these cases, the motivational process appears to gain access to value signals bypassing the DLPFC-CD pathway. A plausible network for the reward size process is prefronto-basal ganglia projections from OFC to rmCD/ventral striatum and/or VP (*Haber et al., 2006*), since ablation or inactivation of these related areas abolished the normal relationship between reward size and error rate in the reward-size task (*Simmons et al., 2010*; *Nagai et al., 2016*; *Fujimoto et al., 2019*). Therefore, our findings, together with our previous results, support the concept that incentive motivation is processed through the prefronto-basal ganglia circuit in accordance with certain topographic organization (*Balleine et al., 2007*; *Haber and Knutson, 2010*). Our findings additionally provide evidence that defines a specific role of dCDh in incentive motivation, as dCDh signals the integrated multi-dimensional factors and contributes to computation of the motivational value.

Our findings may also have some clinical relevance. Dysregulation of normal temporal discounting is associated with increased impulsive behavior. Impulsive behavior and preference are often manifested in patients with psychiatric disorders, including depression, schizophrenia, bipolar disorders, obsessive–compulsive disorders, and substance use disorders (*Pulcu et al., 2014*; *Amlung et al., 2019*). Human imaging studies have revealed the structural and functional connectivity between DLPFC and the striatum with the individual differences in temporal discounting (*van den Bos et al., 2014*; *van den Bos et al., 2015*). Since silencing dCDh did not induce impulsivity (steepened temporal discounting or facilitating reaction was not observed), it could be difficult in the present study to address the link between dCDh activity and mechanisms underlying impulsivity. Nevertheless, our findings may provide a framework to elucidate dysregulation of motivational systems in impulsive individuals with psychiatric disorders.

In summary, our work indicates that dCDh neurons encode, at a single-neuron level, temporally discounted values of forthcoming rewards without reflecting any internal state alteration. These signals are likely to be used in downstream brain structures for formulating motivation of action especially when multi-dimensional factors have to be jointly evaluated.

## Materials and methods

### Subjects

Ten male rhesus macaque monkeys (5–11 kg) were used in this study. Of these, three (BI, FG, and ST) were also used for recording, and one (ST) and two (BI and RI) for chemogenetic and pharmacological inactivation experiments, respectively. All surgical and experimental procedures were approved by the National Institutes for Quantum and Radiological Science and Technology (11-1038-11) and by the Animal Care and Use Committee of the National Institute of Mental Health (Annual Report ZIAMH002619), and were in accordance with the Institute of Laboratory Animal Research *Guide for the Care and Use of Laboratory Animals*.

### Behavioral tasks

The monkeys squatted on a primate chair inside a dark, sound-attenuated, and electrically shielded room. A touch-sensitive bar was mounted on the chair. Visual stimuli were displayed on a computer video monitor in front of the animal. Behavioral control and data acquisition were performed using a real-time experimentation system (REX) (*Hays et al., 1982*). Neurobehavioral Systems Presentation software was used to display visual stimuli (Neurobehavioral Systems).

All monkeys were trained and tested with the delayed reward task (*Figure 1A, B*; *Minamimoto et al., 2009*). In each of the trials, the monkey worked for one of six combinations of reward size and delay. Every trial had the same requirement for obtaining the reward: releasing the bar when a colored spot changed from red to green. Trials began when the monkey touched the bar at the front of the chair. A visual cue and a red spot (wait signal) sequentially appeared in the center of the monitor with a 0.1 s interval. After a variable interval, the red target turned green (go signal). If the monkey released the bar between 0.2 and 1 s after this go signal, the trial was considered correct and the spot turned blue (correct signal). A liquid, either small (one drop, ca. 0.1 mL) or large reward (three drops, except for monkey BI, four drops), was delivered immediately (0.3 ± 0.1 s) or with an additional delay of either 3.3 ± 0.6 s or 6.9 ± 1.2 s after correct release of the bar. Each combination of reward size and delay was chosen with equal probability and independently of the preceding reward condition. An inter-trial interval (ITI) of 1 s was enforced before allowing the next trial to begin. We used a fixed ITI instead of adjusted ITIs with post-reward delays (for example *Blanchard et al., 2013*) because monkeys are insensitive to post-reward delays in our tasks (see Figure 3 in *Minamimoto et al., 2009*). Anticipatory bar releases (before or no later than 0.2 s after the appearance of the go signal) and failures to release the bar within 1 s after the appearance of the go signal were counted as errors. In error trials, the trial was terminated immediately, all visual stimuli disappeared and, following ITI, the trial was repeated, that is, the reward size/delay combination remained the same as in the error trial.

In the behavioral experiment, the visual cue indicated a unique combination of reward size and delay. Two sets of cues were used: a stripe set (for nine monkeys except for BI) and an image set (for monkey BI) (*Figure 1B*). Prior to the behavioral experiment, all monkeys had been trained to perform color discrimination trials in a cued multi-trial reward schedule task for more than 3 months followed by learning of each task for 1–3 months. We collected behavioral data with the delayed reward task for 5–25 daily testing sessions. Each session ended when the monkey would no longer initiate a new trial.

Two monkeys (RI and ST) were also tested with the reward-size task, in which the reward was always delivered immediately (0.3 ± 0.1 s), but the size of the reward (one, two, four, and eight drops) varied and was assigned by unique cue (*Figure 8A*; *Minamimoto et al., 2009*). The sequence and timing of events were the same as those in the delayed reward task.

## Surgery

After behavioral training, magnetic resonance (MR) images at 1.5T (monkey FG) and 7T (monkeys BI, RI, and ST) were obtained under anesthesia (intravenous infusion of propofol 0.2–0.6 mg/kg/min or pentobarbital sodium 15–30 mg/kg) to determine the position of the recording or local injection. After obtaining each MR image, a surgical procedure was carried out under general isoflurane anesthesia (1–2%) to implant chambers for unit recording and/or chemical inactivation. For monkeys BI and FG, we implanted a rectangle chamber (22 × 22 mm ID; KDS Ltd.) from vertical in the coronal plane aiming for the bilateral CD. We implanted one or two cylinder chambers (19 mm ID; Crist Instrument Co., Inc) angled 10° or 20° from vertical in the coronal plane targeting the right or bilateral CD for monkeys ST and RI, respectively. Based on measurements made from the MR images, the centers of the chambers were placed to target the CD near the anterior commissure. A post for head fixation during data collection was also implanted.

## Recording neuronal activity and mapping recording location

Single-unit activity was recorded (51, 31, and 68 from monkeys BI, FG, and ST, respectively), while monkeys performed the delayed reward task in a block usually consisting of 120 trials. Action potentials of single neurons were recorded from the left CD using epoxy-coated 1.1–1.5 MΩ tungsten microelectrodes (Microprobes for Life Science; 1.1–1.5 MΩ at 1 kHz) or glass-coated 1.0 MΩ tungsten microelectrodes (Alpha Omega Engineering Ltd). A guide tube was inserted through the grid hole in the implanted recording chamber into the brain, and the electrodes were advanced through the guide tube by means of a micromanipulator (Narishige MO-97A or Alpha Omega EPS). Spike sorting to isolate single-neuron discharges was performed with a time-window algorithm (TDT-RZ2, Tucker Davis Technologies) or custom-made software written in LabVIEW (National Instruments). Striatal neuronal activities were classified into two subtypes: presumed projection neurons and

tonically active neurons (TANs, presumed cholinergic interneurons) based on their spontaneous discharge rates and action potential waveforms, as previously described (*Yamada et al., 2016*). We exclusively examined the activity of the presumed projection neurons, which are characterized as having a low spontaneous discharge rate (<2 spikes/s) outside the task context and exhibiting phasic discharges in relation to one or more behavioral task events. The activity of TANs recorded from the CD of monkeys performing a similar task was reported in a previous study (*Falcone et al., 2019*). The timing of action potentials was recorded together with all task events at millisecond precision. In the inactivation study, eye movements were monitored for corneal reflection of an infrared light beam through a video camera at a sampling rate of 120 Hz (i_rec, http://staff.aist.go.jp/k.matsuda/eye/).

To confirm the recording location, MR or CT (3D Accuitomo 170, J.MORITA CO.) images were acquired with a tungsten microelectrode (*Figure 1D*). Recording sites extended from 2 mm anterior to the anterior commissure (AC) to 3 mm posterior to the AC for monkey BI, from 4 mm anterior to the AC to 3 mm posterior to the AC for monkey FG, and from 3 mm anterior to the AC to 2 mm posterior to the AC for monkey ST.

## Chemogenetic inactivation

One monkey (ST) received bilateral injections of an adeno-associated virus vector (AAV2.1-hSyn-hM4Di-IRES-AcGFP; 3 µL/site; $4.7 \times 10^{13}$ particles/mL; *Kimura et al., 2021*) at two locations into each side of the CD. The injection procedure was as described previously (*Nagai et al., 2016*). Forty-nine days post-vector injection, the monkey underwent a PET scan with [$^{11}$C]DCZ to visualize in vivo hM4Di expression. Chemogenetic silencing was achieved by intramuscular injection (i.m.) with a DREADD-selective agonist, DCZ (HY-42110, MedChemExpress; 0.1 mg/kg). DCZ was dissolved in 2% dimethyl sulfoxide (DMSO) in saline to a final volume of 0.65 ml. DCZ solution or vehicle (as control) was administered intramuscularly. Five to 10 min following administration, the animal was allowed to start performing the tasks, which continued for 100 min. Based on a previous study, chemogenetic silencing would be effective for 15–120 min after DCZ administration. We performed at most one inactivation study per week. Note that we verified that the DCZ administration (0.1 mg/kg, i.m.) does not cause any significant motivational/motor impairments or alteration of the incentive effect of the performance of reward-size task in monkeys without expressing DREADDs (n = 3) (*Nagai et al., 2020*). Detailed protocols for PET imaging were described elsewhere (*Nagai et al., 2020*).

## Pharmacological inactivation

To inactivate neuronal activity, we injected a GABA$_A$ receptor agonist, muscimol (M1523, Sigma-Aldrich), locally into the bilateral CD of monkeys BI and RI. We used two stainless steel injection cannulae inserted into the CD (O.D. 350 µm; BRC Inc, Japan), one in each hemisphere. Each cannula was connected to a 5 µl microsyringe (Hamilton, #7105KH) via polyethylene tubing. These cannulae were advanced through the guide tube by means of an oil-drive micromanipulator. Muscimol (4 µg/1 µL saline) was injected at a rate of 0.2 µL/min by auto-injector (Legato210, KD Scientific Inc) for a total volume of 3 µL in each side. Soon after the injection was completed, the animal was allowed to start performing the tasks, which continued for 100 min. We performed at most one inactivation study per week. For control, we injected saline at other times using the same parameters as those used for muscimol. At the end of each session, a CT scan was conducted to visualize the injection cannulae in relation to the chambers and skull. The CT images were overlaid on MR images by using PMOD and VirtualPlace (Canon Medical Solutions Corp) image analysis software to assist in identifying the injection sites (*Figures 1D* and *7A*). We plotted the injection sites based on the estimate of the liquid diffusion range (4 mm diameter) reported previously (*Yoshida et al., 1991*; *Martin and Ghez, 1999*).

## Data analysis

The R statistical computing environment (*R Development Core Team, 2004*) was used for all data analyses.

## Behavioral data analysis

Error rates in task performance were calculated by dividing the total number of errors by the total number of trials for each reward condition and then averaged across all sessions. The average error rates in the delayed reward task were fitted with the inverse function of reward size with hyperbolic (*Equation 2*) or that with exponential temporal discounting (*Minamimoto et al., 2009*) as follows:

$$E = \frac{e^{-kD}}{aR} \tag{3}$$

We fitted these two models to the data with least-squares minimization using 'optim' function in R and compared the models by leave-one-out cross-validation as described previously (*Minamimoto et al., 2009*).

To examine the effects of satiation, we divided each session into quartiles based on normalized cumulative reward, $R_{cum}$, which was 0.125, 0.375, 0.625, and 0.875 for the first through fourth quartiles, respectively. We fitted the error rates in the delayed reward task obtained from each monkey and the average data across monkeys to the following model:

$$E = \frac{1 + kD}{aR \times F(R_{cum})} \tag{4}$$

where the satiation effect, $F(R_{cum})$, as the reward value was exponentially decaying in $R_{cum}$ at a constant λ (*Minamimoto et al., 2012a*):

$$F(R_{cum}) = e^{-\lambda R_{cum}} \tag{5}$$

For modeling satiation effects of the error rates in reward-size task, we used an inverse model integrating satiation effect (*Equation 5*), as follows:

$$E = \frac{1}{aR \times F(R_{cum})} \tag{6}$$

We also applied conventional ANOVA modeling to the behavioral data. The proportional behavioral data were transformed using the variance stabilizing arcsine transformation before hypothesis testing (*Zar, 2010*).

The trial initiation time was defined as the duration from the reward of previous trial to the time of lever grip to begin a trial, as a measure of motivation to start a trial. We compared the average trial initiation time in the first and second halves of the daily session.

Significance of deviation from baseline data was examined by means of the parametric bootstrapping method (n = 20,000). We first constructed distribution of the sum of squared residuals (SSR) of the best fit of the model to the averaged resampled error rates (n = 5 or 4 sessions for delayed reward and reward-size task, respectively) from the pooled sample in baseline conditions in each subject. For this analysis, we used normalized error rates by the maximum error rates among reward conditions in each session to remove variance across sessions. p-values for deviation from the distribution were obtained for SSR of the best fit of the model-to-test data (control or inactivation).

The Brunner–Munzel test was used as non-parametric analysis for median value with Bonferroni correction (*Hui et al., 2008*).

## Neuronal data analysis

Only neuronal data from correct trials were used for the analyses. For each neuron, we collected data from 20 to 30 correct trials for each combination of reward-size-and-delay duration, a total of 120–180 successful trials. For each neuron, we first determined the start and end of event-related responses by using a series of $\chi^2$ tests (*Ravel and Richmond, 2006*). The background window was defined as the activity between 500 and 0 ms before cue onset. The test window spanned 100 ms for cue responses, and it moved in 10 ms increments, from 0 to 1500 ms, after cue appearance. For bar-release responses, the 100 ms test window moved from 300 ms before to 300 ms after bar release. For reward-related responses, the 100 ms test window moved from 0 to 500 ms after reward appearance. For each 100 ms test window, an $\chi^2$ test was used to determine whether the proportions of filled to empty 1 ms bins in the 100 ms test interval were significantly different from the

proportion in the 500 ms background window. Start of the response was taken to be the middle of the first of four consecutive 100 ms test intervals showing a significant difference ($p<0.05$) in spike count between the test and background window. End of the response was defined as the middle of the last window showing a significant difference. Duration of the response was defined as the difference between the start and end of the response. The procedure worked well for all tested neurons, yielding latencies that matched those we would have chosen by visual inspection. A neuron was classified as responsive to the three events when a significant response could be detected in at least five consecutive windows.

To quantify the influence of temporal discounting of reward value on the response, we applied linear regression analysis. For each significant response, firing rates ($Y$) were fitted by the following linear regression model:

$$Y = \beta_0 + \beta_V DV \tag{7}$$

where $\beta_V$ is the regression coefficient and $\beta_0$ is the intercept, and $DV$ is the temporally discounted value formulated by a hyperbolic function (*Equation 1*; *Mazur, 1984*; *Mazur, 2001*; *Green and Myerson, 2004*). The effect of $DV$ was compared with that of delay and reward size information on the response by the following multiple linear regression model:

$$Y = \beta_0 + \beta_{delay}D + \beta_{size}R + \beta_{DV}DV \tag{8}$$

where $D$ and $R$ are delay duration and reward size, respectively, $\beta_{delay}$, $\beta_{size}$, and $\beta_{DV}$ are the regression coefficients, and $\beta_0$ is the intercept. Another linear regression analysis was performed to quantify the influence of temporal discounting of reward value and satiation on the response, as follows:

$$Y = \beta_0 + \beta_{DV}DV + \beta_{Rcum}R_{cum} \tag{9}$$

where $DV$ and $R_{cum}$ are the temporally discounted value and cumulative reward, respectively, $\beta_{DV}$ and $\beta_{Rcum}$ are the regression coefficients, and $\beta_0$ is the intercept.

To examine whether DV-coding neurons differentially behave between correct and error trials, we performed LMMs (*Bates et al., 2015*), in which there is mixed effect of trial completion (correct/error) on slope and/or intercept. Four models were nested to consider the presence or absence of random effects (*Figure 4—figure supplement 1*). We applied the LMM analysis on DV-coding neurons recorded in a session in which the monkeys made at least three error trials. The best model was selected based on BIC.

## Acknowledgements

We thank George Dold, Keiji Matsuda, Risa Suma, Tomoni Kokufuta, Anzu Maruyama, Jun Kamei, Yuichi Matsuda, Ryuji Yamaguchi, Yoshio Sugii, Maki Fujiwara, and Mayuko Nakano for their technical assistance.

## Additional information

### Funding

| Funder | Grant reference number | Author |
| --- | --- | --- |
| Japan Society for the Promotion of Science | JP18H04037 | Takafumi Minamimoto |
| Japan Society for the Promotion of Science | JP20H05955 | Takafumi Minamimoto |
| Japan Agency for Medical Research and Development | JP20dm0107146 | Takafumi Minamimoto |
| Japan Agency for Medical Research and Development | JP20dm0207077 | Masahiko Takada |
| Japan Agency for Medical Research and Development | JP20dm0307021 | Ken-ichi Inoue |

| National Institute of Mental Health | ZIAMH-2619 | Barry J Richmond |
| Primate Research Institute, Kyoto University | 2020-A-6 | Takafumi Minamimoto |

The funders had no role in study design, data collection and interpretation, or the decision to submit the work for publication.

## Author contributions

Yukiko Hori, Formal analysis, Investigation, Visualization, Writing - original draft, Writing - review and editing; Koki Mimura, Formal analysis, Visualization, Writing - review and editing; Yuji Nagai, Formal analysis, Investigation, Visualization, Writing - review and editing; Atsushi Fujimoto, Visualization, Writing - review and editing; Kei Oyama, Formal analysis, Writing - review and editing; Erika Kikuchi, Formal analysis, Investigation; Ken-ichi Inoue, Resources, Funding acquisition, Writing - review and editing; Masahiko Takada, Resources, Supervision, Funding acquisition, Writing - review and editing; Tetsuya Suhara, Supervision, Writing - review and editing; Barry J Richmond, Supervision, Funding acquisition, Writing - review and editing; Takafumi Minamimoto, Conceptualization, Formal analysis, Supervision, Funding acquisition, Investigation, Visualization, Writing - original draft, Project administration, Writing - review and editing

## Author ORCIDs

Yukiko Hori http://orcid.org/0000-0003-1023-9587
Yuji Nagai http://orcid.org/0000-0001-7005-0749
Atsushi Fujimoto http://orcid.org/0000-0002-1621-2003
Barry J Richmond http://orcid.org/0000-0002-8234-1540
Takafumi Minamimoto https://orcid.org/0000-0003-4305-0174

## Ethics

Animal experimentation: All surgical and experimental procedures were approved by the National Institutes for Quantum and Radiological Science and Technology (11-1038-11) and by the Animal Care and Use Committee of the National Institute of Mental Health (Annual Report ZIAMH002619), and were in accordance with the Institute of Laboratory Animal Research Guide for the Care and Use of Laboratory Animals.

## Decision letter and Author response

Decision letter https://doi.org/10.7554/eLife.61248.sa1
Author response https://doi.org/10.7554/eLife.61248.sa2

# Additional files

## Supplementary files

- Transparent reporting form

## Data availability

Source data to reproduce the main results of the paper presented in Figures 1, 5, 7 and 8 are provided.

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
