## [Decision Letter]

**Acceptance summary:**

Hori and colleagues tested how neurons in the head of caudate nucleus (dCDh) integrate temporally-discounted reward value and secondarily, how environmental cues may be encoded by this region in action selection. Authors conducted single-unit recordings and pharmacological/chemogenetic inactivations in dCDh in monkeys performing a cue-based reward task. In a revision, the authors have provided additional analyses such as eye gaze and error timings to further corroborate their conclusions for the role of dCdh in integrating time and reward size information in action selection.

**Decision letter after peer review:**

Thank you for submitting your article "Single caudate neurons encode temporally discounted value for formulating motivation for action" for consideration by *eLife*. Your article has been reviewed by 3 peer reviewers, and the evaluation has been overseen by Alicia Izquierdo as the Reviewing Editor and Timothy Behrens as the Senior Editor. The following individuals involved in review of your submission have agreed to reveal their identity: Emmanuel Procyk (Reviewer #1); Ilya E Monosov (Reviewer #3).

The reviewers have discussed the reviews with one another and I have drafted this decision that summarizes the needed changes, to help you prepare a revised submission.

Summary:

Hori and colleagues tested how neurons in the head of caudate nucleus (dCDh) integrate temporally-discounted reward value (DV) and secondarily, how environmental cues may be encoded by this region in action selection. Authors conducted single-unit recordings and pharmacological/chemogenetic inactivations in dCDh in monkeys performing a cue-based reward task. Inactivation validations were provided by microPET binding for DZ and CT scan for muscimol tracks. Overall, reviewers were favorable about this work, and request no additional experiments. However, they identified a need for broader consideration of alternative explanations and additional analyses.

Essential revisions:

A major theme from reviews centered on the need to elaborate on, and better account for, alternative explanations for the role of dCDh. Indeed, there is a rich literature on the basal ganglia in transforming value to action, credit assignment, controlling eye gaze, keeping time, etc. Thus, authors should more thoroughly consider other functions of the striatum in their revision. Additional analyses are being requested and should be included in new or existing Figures: account for the impact of satiety (time on task) and its characterization following dCDh inactivation, provide more information of error types and timing of errors, conduct an analysis of gaze (and time courses of gaze), average time courses of non-DV neuron activity, and also more information about learning stages, accounting for the order of task. Authors should also provide an analysis demonstrating how DV decoding differs in completed vs. aborted trials because this would test directly the decision to act or not. Similarly, authors should provide reaction time effects during inactivation (to include trial initiation, anticipatory behaviors, etc.). These new analyses are essential to bolster a claim of some specificity of dCDh function. These are elaborated on below:

1) The dissociation between DV integration and the satiety (time on task) effect is particularly interesting and very relevant, but a more precise characterization of the consequence of dCDh inactivation should also be possible.

2) A more detailed investigations of errors, and in particular error timings (release time in a trial relative to the different events) could be informed better on the sources of errors. Similarly, the proportion of different error types either (early release vs failure to release). Such timing and proportion analyses might also provide clues on inactivation effects. Theoretically, following the current interpretation, trial aborts should peak at the time of DV encoding.

3) Anterior dorsal regions of the caudate are known to control gaze. In fact, FEF converges on similar regions in the caudate as dlPFC, and both dlPFC and FEF carry some oculomotor signals (spatial signals for example related to saccades), as does the caudate head itself. The authors state that the differences in "value-related" attributes of cues did not impact gaze, but this is not carefully analyzed. What were the time courses of gaze? Might the caudate impact gaze at different epochs of the trial differentially? In other words, could the complexity of gaze behavior mask the relationship of gaze and neural firing rates? Big picture reasons for these questions: it is important to consider this issue because the key claim is that dCDh neurons are encoding motivational (abstract) variables not motor/gaze variables.

4) Provide an account for what non-DV neurons are doing and plot average time courses for all groups of neurons. If possible, plot these and DV neurons' activity also aligned on saccades (gaze shifts to the cues).

5) One may propose an alternative interpretation, that dCDh inactivation perturbs the integration of learned visual cues that have complex associations with reward contingencies, or for which the association is difficult to learn. The authors show interesting control data from a condition with only 4 cues for which there is no effect of inactivation on error rate (according to reward size or time in session). This controls at least for a pure cue recognition/perception deficit. But the learning of the cue-reward size/delay might be a problem. So, could the authors provide information on the learning stages for the different animals, for the main and control tasks, and in what order these tasks have been learned? This actually might be a follow-up point to the previous- error types time distributions and how they inform the causes of errors.

6) For inactivation/disruption – show reaction time effects. Given the complexity of temporal discounting, provide analyses to show the effects of disruption on other task related behaviors. Consider basic measures such as trial initiation, anticipatory behaviors, etc. This will greatly strengthen claims of the "specificity" of effects.

7) The authors mentioned that the activity correlating with DV is related to the process mediating outcome prediction and further the decision to act or not. Indeed, it would be interesting to see how the DV decoding behaves in completed versus aborted trials, to check whether there is any part of the variance explained by the decision to quit the current trial, or not.

A second category of major critiques related to the role of dlPFC, and anatomical placement. All 3 reviewers suggest to downplay the idea that dlPFC is a crucial locus for delay discounting. dlPFC is a heterogenous structure and the over-emphasis in its modulating activity in dCDh (and DV) should be tempered. Relatedly, the tracing experiment for dlPFC projections to caudate (not performed in the monkeys as the inactivation) was deemed redundant and unnecessary, and could be removed. Additionally a note that the location of recorded cells appears more dorsal and medial than the inactivation sites, is worthy of discussion. There are also no data of recordings performed in the presence of inactivation. If authors have these data, it would be good to include them.

---

## [Author Response]

Revisions for this paper:A major theme from reviews centered on the need to elaborate on, and better account for, alternative explanations for the role of dCDh. Indeed, there is a rich literature on the basal ganglia in transforming value to action, credit assignment, controlling eye gaze, keeping time, etc. Thus, authors should more thoroughly consider other functions of the striatum in their revision. Additional analyses are being requested and should be included in new or existing Figures: account for the impact of satiety (time on task) and its characterization following dCDh inactivation, provide more information of error types and timing of errors, conduct an analysis of gaze (and time courses of gaze), average time courses of non-DV neuron activity, and also more information about learning stages, accounting for the order of task. Authors should also provide an analysis demonstrating how DV decoding differs in completed vs. aborted trials because this would test directly the decision to act or not. Similarly, authors should provide reaction time effects during inactivation (to include trial initiation, anticipatory behaviors, etc.). These new analyses are essential to bolster a claim of some specificity of dCDh function. These are elaborated on below:1) The dissociation between DV integration and the satiety (time on task) effect is particularly interesting and very relevant, but a more precise characterization of the consequence of dCDh inactivation should also be possible.

According to the reviewers’ advice, we newly provide plots of how error rates change along with satiation in the dCDh inactivation experiments (Figure 7—figure supplement 4). In the inactivation condition, the satiation effect was still visible in two of three monkeys but was disrupted in monkey ST. Based on these results, we cannot exclude the possibility that the inactivation of dCDh impaired both of two underlying processes: drive and temporal discounting. However, the results shown in Figure 8 indicate that motivation based on drive and its integration with prediction for reward size was normal in two monkeys including monkey ST, and therefore we conclude that “dCDh activity is specifically involved in the motivational value computation based on delay discounting, rather than general motivational process based on the integration of incentives and drive.”

2) A more detailed investigations of errors, and in particular error timings (release time in a trial relative to the different events) could be informed better on the sources of errors. Similarly, the proportion of different error types either (early release vs failure to release). Such timing and proportion analyses might also provide clues on inactivation effects. Theoretically, following the current interpretation, trial aborts should peak at the time of DV encoding.

We provide the proportion and timing of early/late error for all monkeys in Figure 1—figure supplement 1AB. They were not uniform but rather distinctive for each subject. The proportion of early errors differed across monkeys, but it was relatively consistent within each monkey. Most monkeys exhibited a pattern in which early errors increased over time, reaching a peak at about 0.7s or 1.8 s after cue onset; only one monkey (monkey TM) showed an increase in early errors immediately after cue onset. This was indeed comparable to the temporal dynamics of neuronal DV encoding (Figure 5). This suggests that early errors occurred primarily due to insufficient motivation rather than impulsive release of lever. In addition, the late releases did not always occur immediately after the end of the 1s-response window, suggesting that they were not due to extensions of slow reaction. Collectively, these results support the interpretation that errors are caused by insufficient motivation to respond correctly based on DV computation.

We report these data in the Results as follows:

“The proportion of early errors differed across monkeys, but was relatively consistent within each monkey (Figure 1—figure supplement 1A). Nine of ten monkeys exhibited a pattern in which early errors increased over time, reaching a peak at about 0.7s or 1.8 s after cue onset, while only one monkey (monkey TM) showed an increase in early errors immediately after cue onset. These results suggest that early errors were not rejection responses, but rather the consequence of insufficient motivation to make the correct response. In addition, the late releases did not always occur immediately after the end of the 1s-response window, suggesting that they were not due to extensions of slow reaction (Figure 1—figure supplement 1B). These results also support the interpretation that errors are caused by insufficient motivation to respond correctly.”

We also conducted analysis on type of error in inactivation session and reported as follows:

“Type of error (i.e., releasing too early or too late) was unaffected by inactivation (main effect of treatment, *F*_(1,26)_ = 1.07, *p* = 0.31).”

3) Anterior dorsal regions of the caudate are known to control gaze. In fact, FEF converges on similar regions in the caudate as dlPFC, and both dlPFC and FEF carry some oculomotor signals (spatial signals for example related to saccades), as does the caudate head itself. The authors state that the differences in "value-related" attributes of cues did not impact gaze, but this is not carefully analyzed. What were the time courses of gaze? Might the caudate impact gaze at different epochs of the trial differentially? In other words, could the complexity of gaze behavior mask the relationship of gaze and neural firing rates? Big picture reasons for these questions: it is important to consider this issue because the key claim is that dCDh neurons are encoding motivational (abstract) variables not motor/gaze variables.

Because we monitored eye movements only during inactivation and control sessions, but not during the recording, we are unable to provide direct evidence for the relationship between firing rate and gaze. Instead, we provided a time course of the probability of eye position within the cue area aligned by CUE and GO onset in Figure 1—figure supplement 2B. Although we did not require the monkeys to fixate on the cue, they continued to look at the cue with similar temporal dynamics from onset to go, regardless of the reward condition. In addition, our data in Figure 1—figure supplement 2A also shows a similar tendency of eye position during cue period, together supporting that the attribution of cue value does not have a significant or systematic effect on eye position. We propose to interpret our results as follows:

"The DV relation was not likely to be a direct reflection of the eye movement or gaze variables, since the monkeys tended to look at cue location from cue to go signal regardless of rewarding condition (Figure 1—figure supplement 2).”

4) Provide an account for what non-DV neurons are doing and plot average time courses for all groups of neurons. If possible, plot these and DV neurons' activity also aligned on saccades (gaze shifts to the cues).

According to the reviewers’ suggestion, we plotted the average time course for non-DV coding neurons (n = 73) and DV-coding neurons (n = 27) (Figure 5EF). In non-DV coding neurons, discharge increased rapidly after CUE onset, then decreasing to the baseline level around the time of release (Figure 5EF, black). By contrast, in DV-coding neurons, discharges increased just before the appearance of cue, reached a peak about 1s after the cue onset, and then decreased after the release (Figure 5EF, red). As mentioned above, since we monitored eye movements only during inactivation experiments, we are not able to provide a plot of neuronal activities aligned on saccades. Because the monkeys tended to look at the cue position before cue onset and after release, the activity of non-DV coding neurons appeared to reflect mainly visual response, but was unlikely to reflect only eye movement. These different response profiles suggest that non-DV neurons might contribute to detection of cue appearance.

We added the time course plots and the possible contribution of non-DV neurons in the Results as follows:

“Non-DV coding neurons, on the other hand, did not change the effect size from 0 during the cue period, whereas it increased after the bar release (black curve, Figure 5C and D). Comparing the normalized activity of these two populations, whereas DV coding neurons showed an increase in activity toward the bar release, non-DV coding neurons showed a marked transient response to the cue (Figure 5E and F). Given that the monkeys tended to look at the cue location during cue period (Figure 1—figure supplement 2B), the activity of non-DV coding neurons appeared to largely reflect visual response, but was unlikely to be evoked by eye movement. This suggests that non-DV coding neurons might have a role in detecting cue appearance.”

5) One may propose an alternative interpretation, that dCDh inactivation perturbs the integration of learned visual cues that have complex associations with reward contingencies, or for which the association is difficult to learn. The authors show interesting control data from a condition with only 4 cues for which there is no effect of inactivation on error rate (according to reward size or time in session). This controls at least for a pure cue recognition/perception deficit. But the learning of the cue-reward size/delay might be a problem. So, could the authors provide information on the learning stages for the different animals, for the main and control tasks, and in what order these tasks have been learned? This actually might be a follow-up point to the previous- error types time distributions and how they inform the causes of errors.

Thank you for pointing out this issue. The training order for two tasks was balanced between monkeys RI and ST: the delayed reward task was introduced first in monkey RI but the reward size task was first in monkey ST. We added Figure 8—figure supplement 1, displaying the performance of these tasks during the learning stage in the two monkeys. They initially showed error rate patterns independent of outcome value (gray), but following 2 to 2.5 months of training, they showed stable and reasonable patterns (yellow). There was no clear difference between the tasks in the number of days necessary to reach the criterion of ‘learned’ (stable error rate patterns in 5 consecutive days while providing better fit for the model R^2^ > 0.9). These results suggest that the delayed reward task was not a particularly difficult task to establish the cue-outcome association compared to the reward size task. We added the above argument by referring to the supplement in the Results section as follows:

“The lack of change in the error rate pattern in the reward size task could be attributed to the relative ease of associating cues with outcome compared to the delayed reward task. However, no clear difference was evident between the two tasks in establishing the cue-outcome relationship as judged by the behavior during the training period (Figure 8—figure supplement 1).”

6) For inactivation/disruption – show reaction time effects. Given the complexity of temporal discounting, provide analyses to show the effects of disruption on other task related behaviors. Consider basic measures such as trial initiation, anticipatory behaviors, etc. This will greatly strengthen claims of the "specificity" of effects.

According to the reviewers’ advice, we provided (a) reaction time data and performed additional analyses on (b) trial initiation and (c) anticipatory behaviors. We thank the reviewers for their constructive comments, as these additional data certainly strengthen our claim that dCDh inactivation is specific to temporal discounting.

a) Reaction time:

We provided the results of the effect of dCDh inactivation on reaction time (Figure 7—figure supplement 1). There was no significant effect of treatment on reaction time in two monkeys (two-way ANOVA, effect size of treatment: η^2^ < 0.01, monkeys BI and RI) but a moderate effect of treatment in monkey ST (η^2^ = 0.06). This is reported in Results.

b) Trial initiation:

We examined trial initiation time, which was defined as the duration from the reward of the previous trial to the time of lever grip to begin a trial, as a measure of motivation to start a trial. We previously demonstrated that the trial initiation time was significantly extended in the second half compared to the first half in intact monkeys, which suggested a satiation effect on the computation of motivational value (Fujimoto et al., J Neurosci 2019). Here, we measured the trial initiation time in 3 monkeys in the delayed reward task. Two-way ANOVA revealed significant main effect of session period (1st vs 2nd, F(1,54) = 4.32, p = 0.042), but no effect of dCDh inactivation (1st vs 2nd × treatment, F(1,54) = 0.32, p = 0.57) on the trial initiation time, suggesting that there was no impact of inactivation on satiation effect. We added this in the Results section.

c) Anticipatory behaviors:

The proportion of early errors among total errors is a measure of anticipatory behavior. We confirmed that the type of error (i.e., releasing too early or too late) was unaffected by inactivation (main effect of treatment, F(1,26) = 1.07, p = 0.31).

We also measured the number of times the monkeys released the lever during the delay period. Even though the delay time was not shortened, the monkeys touched and released the bar several times during the delay period. This can be considered another measure of monkeys' expectation of a future reward. We confirmed that this anticipatory behavior during delay was unchanged by treatment (two-way ANOVA, treatment, F(1, 143) = 2.90, p = 0.09). The number of releases depended on reward condition (main effect of reward condition, F(5, 143) = 25.22, p < 0.001), but there was no significant interaction with treatment (F(5, 143) = 0.42 , p = 0.83). We added this in the Results.

7) The authors mentioned that the activity correlating with DV is related to the process mediating outcome prediction and further the decision to act or not. Indeed, it would be interesting to see how the DV decoding behaves in completed versus aborted trials, to check whether there is any part of the variance explained by the decision to quit the current trial, or not.

We appreciate this important advice from the reviewers. We analyzed the effect of error/correct trials on the activity of DV-coding neurons using linear mixed model analysis, implementing mixed effects of slope and/or intercept to explain the data. This analysis was applied to 22 of the 27 DV neurons recorded in a session in which the monkeys made at least three error trials. As shown in the table (Figure 3—figure supplements 1), the majority of DV neurons (17 of 22) behaved differently between in correct and in aborted trials, supporting our interpretation that the activity of these neurons is related to processes that mediate outcome prediction and the decision to act or not. We reported these in the Results by referring to Figure 3—figure supplements 1 and 2, as follows:

“We postulated that the activity of DV-coding neurons may be related to the process mediating outcome prediction and further the decision to act or not. If this is the case, the DV coding neurons should behave differentially between correct and error trials. To test this, we performed linear mixed model (LMM) analysis on 22 of 27 DV-coding neurons recorded in a session in which the monkeys made at least three error trials. We found that the majority of DV-coding neurons (17 of 22) were modulated differentially by DV depending on whether the monkey performed correctly or not (Figure 3—figure supplements 1 and 2), supporting the idea that this population of neurons is involved in motivational processes.”

A second category of major critiques related to the role of dlPFC, and anatomical placement. All 3 reviewers suggest to downplay the idea that dlPFC is a crucial locus for delay discounting. dlPFC is a heterogenous structure and the over-emphasis in its modulating activity in dCDh (and DV) should be tempered. Relatedly, the tracing experiment for dlPFC projections to caudate (not performed in the monkeys as the inactivation) was deemed redundant and unnecessary, and could be removed.

According to the reviewers’ suggestion, we minimized the idea that dlPFC is a crucial locus for delay discounting. Instead, we stated that the basal ganglia are the major candidate for motivational value computation and that dCDh can be a target region for motivational process with temporal discounting, as it receives convergent inputs from multiple prefrontal areas including DLPFC, ACC, and SEF, where neuronal information about size and timing of reward has been reported.

We also omitted the tracing results (original Figure1—figure supplement 2). We reorganized the Summary, Introduction and Discussion accordingly.

Additionally a note that the location of recorded cells appears more dorsal and medial than the inactivation sites, is worthy of discussion. There are also no data of recordings performed in the presence of inactivation. If authors have these data, it would be good to include them.

We have modified the plot of muscimol inactivation sites including the data from two monkeys (Figure 7B) based on the estimated extent of muscimol diffusion, which is 4 mm in diameter according to previous studies (Martin and Ghez, J Neurosci Methods 1999; Yoshida et al., Neurosci Res, 1991). These show that inactivation sites appeared to cover the location of recorded cells (cf. Figure 1D).

We agree with the importance of showing recording data during inactivation, especially since DREADD inactivation is currently under development in monkeys. However, we did not make recordings during inactivation in this study. We have preliminary data of silencing neuronal activity using a similar experimental condition (i.e., using hM4Di-DREADD and DCZ). But this study focused on the neural mechanisms of time discounting, and did not go so far as to address this issue, and we hope to report this in a future study.